# The Establishment and Application of Indirect 3AB-ELISA for the Detection of Antibodies against Senecavirus A

**DOI:** 10.3390/v15040861

**Published:** 2023-03-28

**Authors:** Junfang Yan, Yanni Gao, Jian Li, Minjing Li, Chengyi Guo, Juan Bai, Ping Jiang

**Affiliations:** 1Key Laboratory of Animal Diseases Diagnostic and Immunology, Ministry of Agriculture, MOE International Joint Collaborative Research Laboratory for Animal Health & Food Safety, College of Veterinary Medicine, Nanjing Agricultural University, Nanjing 210095, China; 2Jiangsu Co-Innovation Center for Prevention and Control of Important Animal Infectious Diseases and Zoonoses, Yangzhou 225000, China

**Keywords:** SVA, recombinant 3AB protein, indirect ELISA, serological survey

## Abstract

Senecavirus A (SVA) is an emerging pathogen that negatively affects the pig industry in China. Affected animals present vesicular lesions which are indistinguishable from other vesicular diseases. To date, there is no commercial vaccine that can be used to control SVA infection in China. In this study, recombinant SVA 3AB, 2C, 3C, 3D, L and VP1 proteins are expressed by using a prokaryotic expression system. The kinetics of the presence and levels of SVA antibodies with SVA-inoculated pig serum show that 3AB has the best antigenicity. An indirect enzyme-linked immunosorbent assay (ELISA) is developed with the 3AB protein, exhibiting a sensitivity of 91.3% and no cross-reaction with serum antibodies against PRRSV, CSFV, PRV, PCV2 or O-type FMDV. Given the high sensitivity and specificity of this approach, a nine-year (2014–2022) retrospective and prospective serological study is conducted to determine the epidemiological profile and dynamics of SVA in East China. Although SVA seropositivity declined markedly from 2016 (98.85%) to 2022 (62.40%), SVA transmission continues in China. Consequently, the SVA 3AB-based indirect ELISA has good sensitivity and specificity and is suitable for viral detection, field surveillance and epidemiological studies.

## 1. Introduction

Senecavirus A (SVA), previously designated Seneca Valley virus (SVV), is a member of the genus *Senecavirus* belonging to the Picornaviridae family. SVA was discovered as a serendipitous finding while cultivating adenovirus-5 (Ad5)-based vectors in PER.C6 cells [1]. It is an emerging pathogen that negatively affects the pig industry. To date, SVA has been found in Canada [2], Brazil [3], the United States [4], Colombia [5], Thailand [6] and China [7]. Clinical symptoms include fluid-filled and ruptured vesicles or ulcerative lesions on the snouts and coronary bands of pigs. Lameness, anorexia, lethargy, cutaneous hyperemia and fever are observed in infected pigs. SVA infections are related to vesicular lesions which are indistinguishable from other vesicular diseases, such as foot-and-mouth disease (FMD), swine vesicular disease (SVD) and vesicular stomatitis (VS) [8]. Clinical diagnosis is not sufficient for SVA confirmation.

SVA is a nonenveloped single-stranded RNA virus. The genome has a single open reading frame (ORF) flanked by a 5′ untranslated region (UTR) and a short 3′ UTR followed by a poly (A) tail. Subsequently, ORF is translated into a polyprotein which is cleaved by the viral protease into four structural proteins and seven non-structural proteins, following the standard L-4-3-4 layout of the picornavirus genome (L-VP4-VP2-VP3-VP1-2A-2B-2C-3A-3B-3C-3D) [1,9]. Among them, the capsid protein VP1 is the most immunodominant in the Picornaviridae family [10]. Moreover, non-protective antibodies against non-structural proteins in response to infection are induced [11]. For instance, an indirect ELISA based on the FMDV 3AB protein was established to specifically identify antibodies induced by FMDV infection but not those induced by vaccination [12]. Nevertheless, there are no commercial vaccines to prevent and control SVA infection and the presence of SVA antibodies indicates current or historic infection. Therefore, laboratory diagnoses such as serological assays play a key role during SVA diagnosis. Accurate surveillance of SVA-specific antibodies in pigs would be essential for SVA control.

The advantage of antibody detection assays is the ability to process large numbers of samples in epidemiological surveillance and mass diagnostic programs [13]. Moreover, serological diagnosis has become the most commonly used diagnostic method because of its simplicity, relatively low cost and low requirements for specialized equipment [14]. ELISA, which not only sharply simplifies the detection process but also greatly increases the sensitivity and specialty, is a rapid, effective serological method for evaluating the amount of intact virus in a vaccine [15]. A series of ELISAs have been developed for detecting SVA antibodies. The competitive enzyme-linked immunosorbent assay (cELISA) methodology using a developed SVA VP2 monoclonal antibody (mAb) offers a promising approach for a rapid and convenient serodiagnosis [16]. However, the screening and identification of well-characterized monoclonal antibodies are time-consuming and laborious. Subsequently, a SVA VP1 recombinant protein (rVP1) indirect ELISA was applied to detect the serological response (IgG) to SVA [17]. Nevertheless, few studies have clarified that the antigenic reactivity of SVA VP1 is not the most immunodominant. Recently, an indirect ELISA based on the VP2 epitope (VP2-epitp-ELISA) was developed to detect antibodies directed against SVA [18]. The method has not been utilized to process large numbers of samples in epidemiological surveillance and mass diagnostic programs. As a result, better coated antigens need to be screened and identified in preparation for subsequent large-scale seroepidemiological surveys.

In the present study, a panel of SVA viral proteins were expressed and examined for antigenicity with SVA-positive serum. The kinetics of the presence and levels of SVA antibodies with SVA-inoculated pig serum showed that 3AB had the best antigenicity. The reaction conditions of 3AB indirect ELISA were successively screened and optimized. The established indirect ELISA was sensitive and specific to SVA antibody detection and was applied for SVA antibody surveillance, with 3930 samples collected in East China from 2014 to 2022. A retrospective and prospective serological study revealed that the overall seroprevalence was about 80%, suggesting that SVA is still endemic in East China. Surprisingly, from 2014 the seropositivity rate of SVA in China was relatively high. After 2016, the serum prevalence of SVA gradually decreased. By 2022, the geographical distribution of SVA was significantly different. The serum positive ratio in Shandong and Jiangxi provinces had fallen below 40%. In addition, the detection rate in piglets was about 60%, implying the importance of maternal antibodies. In summary, a retrospective epidemiological investigation based on well-characterized 3AB indirect ELISA elucidated the prevalence trend, the seasonal and geographical distribution of SVA and the differences in SVA infection among herds in East China. The results lay the foundation for the subsequent prevention and control of SVA.

## 2. Materials and Methods

### 2.1. Serum Samples

The serum used to detect the kinetics of antibodies was collected from experimentally SVA-inoculated pigs. A total of 3930 serum samples used for SVA surveillance were collected from 158 pig farms in East China from 2014 to 2022 and stored at −20 °C until use. Furthermore, 54 negative and 46 positive serum samples selected to evaluate the established ELISA were identified by a virus neutralization test (VNT). Porcine reproductive and respiratory syndrome virus (PRRSV), classical swine fever virus (CSFV), pseudorabies virus (PRV), porcine circovirus type 2 (PCV2) and serotype O foot-and-mouth disease virus (O-type FMDV)-positive serum were stored in the lab.

### 2.2. Expression of Recombinant Protein

Viral RNA was extracted from SVA using Trizol reagent (Invitrogen) following the manufacturer’s instructions. The sequence corresponding to the genes was amplified from viral RNA by RT-PCR using oligonucleotide primers designed according to the SVV-CH-SD strain (GenBank: MH779611) (Table 1). The SVA gene fragments were cloned and expressed in *E. coli* as a histidine (His)-tagged recombinant protein. Expression was facilitated by adding 0.5 mM isopropyl-β-D-1-thiogalactoside (IPTG) when the optical density at 600 nm of the culture reached 0.6 [19], and successful expression was examined by sodium dodecyl sulfate-polyacrylamide gel electrophoresis (SDS-PAGE) analysis of cell lysates [20].

### 2.3. Purification of Recombinant Protein

For the purification process, Ni^2+^-NTA affinity chromatography was employed [21]. The column was first equilibrated with 20 mM phosphate buffer (pH 8.0) and 100 mM NaCl, followed by a loading of the soluble fractions obtained above. The elution of bound protein was performed through a linear gradient of 500 mM imidazole using the ÄKTA pure protein purification system (GE Healthcare) [22,23]. The eluted fractions containing the target proteins were subjected to a desalting column for desalination and finally ultrafiltered [24].

The purification of inclusion bodies (IB) was operated as previously described [25,26]. The lysis buffer containing either 6 M guanidine hydrochloride (GuHCl), 8 M urea or 0.3% Sarkosyl (n-lauroyl sarcosinate) was routinely used for IB solubilization. Solubilized proteins were diluted 10 times for a final concentration between 100 and 500 μg/mL in basic refolding buffer for 2 h at room temperature followed by incubating at 4 °C for 18 h. Refolded proteins were then centrifuged at 14,000 RPM for 15 min to remove any aggregates and dialyzed with PBS using 10 kDa ultracentrifugal filters.

### 2.4. SDS-PAGE and Western Blotting

In order to intuitively visualize the expression and purification effect of recombinant proteins, SDS-PAGE was performed. Purified recombinant proteins were subjected to a 12.5% SDS-PAGE by electrophoresis and determined by the dying method with Coomassie brilliant blue. Meanwhile, a Western blot was utilized to identify the specific reaction between the recombinant protein and the primary antibody. Similarly, purified recombinant proteins were subjected to a 12.5% SDS-PAGE by electrophoresis and separated proteins were transferred onto a nitrocellulose filter (NC) membrane. The membranes were blocked for 2 h at room temperature (RT) with 5% nonfat dry milk. After blocking and washing with Tris-buffered saline containing 0.05% Tween 20 (PBST), primary antibody incubation was performed with anti-His monoclonal antibodies (1:5000 dilution) at RT for 2 h. After washing three times with PBST, HRP-labeled goat anti-mouse IgG was added. After a final washing step, chemiluminescence was carried out [27].

### 2.5. ELISA Procedure

For the screening of the best-coated antigen, the standard protocol for indirect ELISA was performed [28,29]. Briefly, ninety-six-well microtiter ELISA plates were coated with a concentration of 2 μg/mL protein in carbonate buffer and incubated overnight at 4 °C. After washing with Tris-buffered saline containing 0.05% Tween 20 (PBST), the plates were blocked with PBST containing 5% nonfat dry milk at 37 °C for 3 h. Subsequently, SVA-positive serum was added to the plates at 37 °C for 60 min. After another washing step, horseradish peroxidase (HRP)-conjugated goat anti-pig IgG (1:10,000 dilution) was added for 45 min incubation at 37 °C. Antibody binding was analyzed by adding 100 μL TMB for 10 min, and then, 2 M H_2_SO_4_ was added to stop the reaction. Finally, the absorbance was measured at OD_450_.

### 2.6. Antibody Kinetics Analysis

Animal experiments were described in detail in our previous research [8]. SVA-inoculated pig serum was collected at seven-day intervals and stored at −20 °C until use. The kinetics of the presence and levels of SVA antibodies with SVA-inoculated pig serum were detected by ELISA based on different viral proteins. The consistency of the conditions across a range of ELISA methods should be considered. All procedures strictly followed the ELISA procedure mentioned above. By comparing the antibody levels of different encoded proteins in the sera of SVA-inoculated pigs, the optimally coated antigens were determined for subsequent serological detection.

### 2.7. Virus Neutralizing Test (VNT)

In order to evaluate the reliability of the established ELISA method, a virus neutralization assay was performed as the gold standard for detection. VNT was performed according to the procedure described by OIE. First, sera were heat inactivated at 56 °C for 60 min. Replicates of two-fold serially diluted sera (50 μL/well, starting from 1/4) were mixed with an equal volume of 100TCID_50_ of SVA and incubated at 37 °C for 60 min [30]. Thereafter, the cell suspension was added to the wells. Plates were incubated for another 3 days at 37 °C in 5% CO_2_. Test results were evaluated using an optical microscope. Titer was read as the highest dilution of serum, resulting in 50% CPE reduction [31]. Sera with neutralizing titers > 1/64 were considered positive.

### 2.8. Retrospective Serological Survey by 3AB ELISA

A retrospective serological survey was exploited to trace the origin of SVA outbreaks and determine the epidemiological profile and dynamics of SVA. First, 3930 serum samples used for SVA surveillance were collected from 158 pig farms in East China from 2014 to 2022. Serological investigation was carried out based on an optimized 3AB indirect ELISA method. The geographical distribution, prevalence trend and seasonal prevalence of the clinical sera were analyzed. At the same time, serological surveillance statistics were organized by growth stage (piglets, nursery pig, fat pig, reserve sow, pregnant sow, farrowing sow and boar). Detection data were presented as S/P values. All statistical analyses were performed using GraphPad Prism 6.0 software.

## 3. Results

### 3.1. Purification and Identification of SVA Protein

Analysis of the bacterial cell lysate by SDS-PAGE and Western blotting revealed a prominent band corresponding to each protein. All recombinant proteins were efficiently expressed by the prokaryotic expression system. The recombinant protein 3AB and VP1 were predominantly in the soluble fraction of the bacterial cell extract and were purified using Ni^2+^ affinity chromatography. 2C, 3C, 3D and L proteins of SVA were expressed in the form of the insoluble fraction. These proteins had a better purification effect (Figure 1A). Western blot analysis showed that these purified proteins were specifically recognized by the antibody against His-tag (Figure 1B).

### 3.2. Antibody Kinetic Analysis

The presence and levels of SVA antibodies in SVA-inoculated pig serum were kinetically analyzed, which indicated that the antibody levels of the 3AB protein were highly immunogenic. Moreover, the persistence of antibodies against 3AB in the serum was considerable (Figure 2A). Although the RNA helicase 2C protein and chymotrypsin-like cysteine protease 3C play a crucial role in innate immune response and apoptosis signaling, antibodies against 2C and 3C in the sera of infected pigs were not the most prominent (Figure 2B,D). The titers of antibodies against viral RNA-dependent polymerase 3D were also much lower than those against the 3AB protein (Figure 2C). In picornavirus, VP1 is characterized as the most antigenic virus-encoding protein. However, our experimental results showed that serum antibody levels against VP1 in infected pigs were relatively low, comparable to those against the leader protein L, and much lower than several other non-structural proteins (Figure 2E,F). Therefore, it was determined that 3AB was used as an ELISA coating antigen to establish a corresponding antibody detection method.

### 3.3. Indirect ELISA for SVA 3AB

A checkerboard titration involving each combination of antigen (purified recombinant SVA 3AB protein) and sera was used to determine the optimal dilutions for use in the indirect ELISA we developed. The antigen and serum were individually diluted. In addition, the antigen encapsulation time, blocking agent and blocking time were successively identified. Subsequently, the incubation time of the serum, the dilution ratio, the incubation time of the secondary antibody and the duration time of TMB were confirmed. The optimal conditions were selected based on the principle that the ratio of the positive control to the negative control (P/N value) was maximal and the OD_450_ of the positive control was as close to 1.0 as possible. Through optimization, ninety-six-well microtiter ELISA plates were coated with a concentration of 1.5 μg/mL protein in carbonate buffer (Figure 3A) and incubated overnight at 4 °C. After washing with PBST, the plates were blocked with 5% nonfat dry milk at 37 °C for 3 h (Figure 3B). Subsequently, SVA-positive serum (1:100 dilution) was added to the plates at 37 °C for 45 min (Figure 3C). After another washing step, horseradish peroxidase-labeled Staphylococcus aureus protein A (HRP-SPA) (1:10,000 dilution) was added for 30 min incubation at 37 °C (Figure 3D,E). Antibody binding was analyzed by adding 100 μL TMB for 10 min (Figure 3F).

In order to exclude the error caused by environmental factors and operator differences, the serum antibody response was expressed as sample-to-positive (S/P) ratios:

S/P ratio = (sample OD_450_ − negative-control mean OD_450_)/(positive-control mean OD_450_ − negative-control mean OD_450_).

The criterion for positive serum was S/P ≥ 0.30, the cut-off value for negative serum was 0.2 and suspicious samples were between the two values. In total, 54 negative serum and 46 positive serum samples were determined by virus neutralization test. Subsequently, the established 3AB ELISA method was used to determine these sera. By comparison, the sensitivity, specificity and coincidence rate of 3AB indirect ELISA were all higher than 90% (Table 2).

### 3.4. Retrospective Serological Survey

Given the high sensitivity and specificity of 3AB ELISA, a nine-year (2014–2022) retrospective and prospective serological study was conducted to determine the epidemiological profile and dynamics of SVA in East China. A total of 3930 serum samples used for SVA surveillance were collected from 158 pig farms (69/158, information integrity). The area of light green in the map represented the sampled provinces (Figure 4A). Overall, the seroprevalence (82.19%) of SVA was generally higher in East China. The year 2016 was considered to be a turning point in the epidemiology of SVA in China, which is consistent with previous reports [32]. In subsequent years, the seropositivity rate showed a downward trend (Figure 4B). The possible reasons for this difference were the evolution of the strain. The SVA strains isolated before 2016 and the strains isolated in Canada and Brazil showed high homology, whereas the strains isolated after 2016 were all more closely related to the US strain. Additionally, in terms of the seasonal distribution of the disease, the serum positive rate was lower in autumn than in other seasons (Figure 4C).

To further compare the differences in SVA infection between different regions, the number of seropositive ratio from 11 provinces was counted (Table 3). Due to the frequent outbreaks of SVA in 2015 and 2016, the total seropositivity rate was above 80% during the investigation. These results showed that there was no significant difference in the prevalence of SVA in different provinces. Additionally, the number of blood samples submitted was unsatisfactory and the data were not representative in Sichuan, Shanxi, Fujian and Anhui provinces. It was encouraging that the epidemic of SVA had begun to slow down. Our results revealed a significant downward trend in SVA transmission in Shandong, Jiangsu and Jiangxi provinces from 2014 to 2022 (Table 4). Most encouragingly, the serum positive ratio in Shandong and Jiangxi provinces had fallen below 40% in 2022. However, the fluctuation of the seroprevalence of SVA in Zhejiang Province should not be underestimated. Moreover, the seropositivity rate of piglets and nursery pigs was around 60% (Figure 5), implying that the offspring was protected against SVA infection by maternal antibodies. Conversely, the exposure rate of sows and boars was extremely high, which greatly enlarged the difficulty of introducing breeding.

## 4. Discussion

SVA is a new pathogen that negatively affects the pig industry in China. The clinical symptoms after pig infection are indistinguishable from blister animal diseases such as foot-and-mouth disease. At present, there are no commercial vaccines at home or abroad [33]. Given that ELISA is a simple, sensitive and convenient serological detection method, many ELISA methods were developed to detect SVA antibodies against SVA. An SVV-specific competitive enzyme-linked immunosorbent assay (cELISA) was developed using BEI-inactivated SVV antigen and a monoclonal antibody (mAb) for serodiagnosis. The cELISA reflects the profile of the indirect ELISA for both the IgM and IgG [16]. Furthermore, the serological response (IgG) to SVA is evaluated by sow and piglet serum samples on an SVA VP1 recombinant protein (rVP1) indirect enzyme-linked immunosorbent assay (ELISA). Seroconversion against SVA in clinically affected and non-clinically affected sows at early stages of the outbreak and maternal SVA antibodies in offspring were detected by rVP1 ELISA [17]. Recently, an indirect ELISA based on the VP2 epitope (VP2-epitp-ELISA) was developed to detect antibodies directed against SVA, and no cross-reaction with positive serum antibodies occurred for other idiopathic vesicular diseases [18]. Although these ELISA methods were well characterized, none of them were extended to mass diagnostic programs.

Non-structural protein 3AB is considered a superior diagnostic antigen due to its association with viral replication or infection. By comparing the antibody kinetics of the encoded protein in the serum of infected pigs, a high antibody titer against SVA 3AB protein was observed. At the same time, the antibody presence against 3AB in the sera of infected pigs had a longer duration (Figure 2A). Based on this characteristic, indirect ELISA based on the 3AB protein could maximize the identification of SVA infection. In addition, compared with the virus neutralization test, 3AB indirect ELISA had a higher sensitivity (91.30%) and specificity (92.59%). Therefore, the method is expected to be further commercially developed and used to easily and quickly detect a large number of clinical sera and carry out an epidemiological investigation.

Retrospective and prospective serological studies of infectious diseases are important to outline the health profile of livestock and represent important tools for the establishment of the time of the entrance of an infectious agent into a country, as well as for determining the risk factors associated with the infection [34]. A previous retrospective serological survey conducted by foreign scholars in Brazil proved that SVV was not present in the major Brazilian pig-producing regions before 2014. However, SVA infections had begun to spread rapidly among pigs of different ages in China in 2015. In order to determine the origin of SVA outbreaks, a nine-year retrospective serological survey based on 3AB ELISA was performed. The results suggested that SVA had a higher titer in pig sera in 2014, indicating that SVA had been present in China before that year. Of course, this phenomenon needs more evidence and corroboration. Furthermore, the SVA strains isolated before 2016 were more closely related to the Canadian strains and the Brazilian strains. However, the strains isolated after 2016 were all more closely related to the US strain. Therefore, 2016 was a turning point for the epidemiology of SVA in China. Our research also reinforced this idea. Before 2016, the seropositive rate of SVA had been quickly boosted but after 2016, the prevalence trend of SVA in China slowed down, indicating that the change in the SVA epidemic might be due to the difference in the virus strains.

To further observe the prevalence of SVA in China, the seroprevalence of 3930 samples from 11 cities was analyzed. In recent years, the serum prevalence of SVA showed a significant downward trend (Figure 4B). In terms of the seasonal distribution, the prevalence rate in autumn was slightly lower (Figure 4C). There was no significant difference in the overall positive rate among different provinces in East China (Table 3). However, the seroprevalence for Jiangsu (75.0%) and Zhejiang (81.6%) provinces was still high, while for Shandong and Jiangxi provinces, it was lower than 40% in 2022 (Table 4). In 2022, the prevalence of SVA varied greatly regionally, possibly due to differences in biosafety awareness among breeders. The specific reason remains unknown and awaits further investigation. Since vesicles and erosions associated with SVA infection were seen mainly in sows and fattening (finisher) pigs [9], our study also calculated the seroprevalence of pigs at different growth stages. The antibody-positive rate of piglets and nursery pigs was relatively low at about 60% (Figure 5). We hypothesized that the maternal antibodies acquired by the piglets from the sows further protected them from SVA infection. It was regrettable that the seroprevalence of fat pigs, reserve sows, pregnant sows, farrowing sows and boar was higher.

In conclusion, the SVA 3AB ELISA antibody detection method was developed successfully with a high degree of sensitivity and specificity. It is expected to be developed into a commercial antibody detection kit. Nine-year retrospective serological data based on 3AB ELISA characterized the current epidemic situation of SVA in China for the first time and will provide constructive guidance for subsequent prevention, diagnosis and treatment.

## Figures and Tables

**Figure 1 viruses-15-00861-f001:**
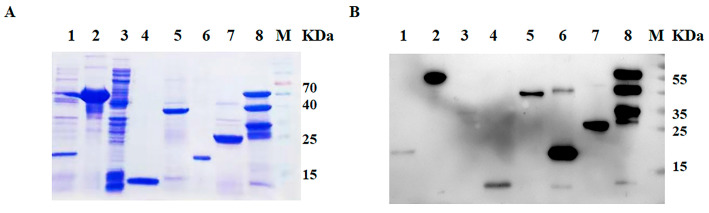
Purification and identification of SVA protein. (**A**) SDS-PAGE. (**B**) Western blot. Lane 1 is the pET-32a (m) control, lane 2 is the VP1 recombinant protein, lane 3 is the pET-28a control, lane 4 is the L recombinant protein, lane 5 is the 2C recombinant protein, lane 6 is the 3AB recombination protein, lane 7 is the 3C recombinant protein and lane 8 is the 3D recombinant protein. M is the protein ladder. The order was the same in (**A**,**B**).

**Figure 2 viruses-15-00861-f002:**
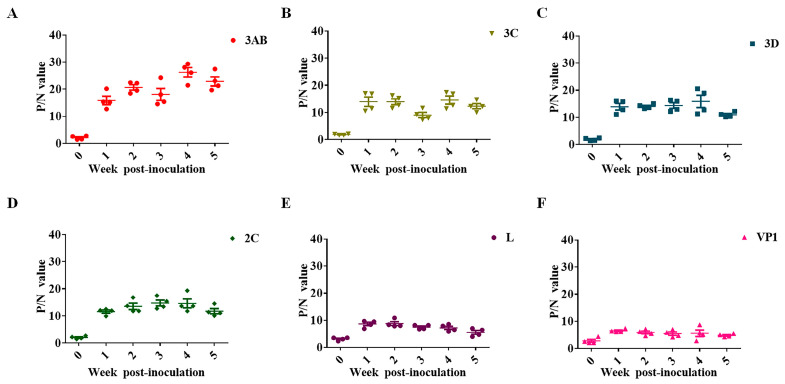
Dynamic analysis of antibodies encoding different viral proteins in SVA-infected swine. (**A**) Antibody kinetics against the SVA 3AB protein. (**B**) Antibody kinetics against the SVA 3C protein. (**C**) Antibody kinetics against the SVA 3D protein. (**D**) Antibody kinetics against the SVA 2C protein. (**E**) Antibody kinetics against the SVA L protein. (**F**) Antibody kinetics against the SVA VP1 protein. P/N value, the ratio of positive control to a negative control.

**Figure 3 viruses-15-00861-f003:**
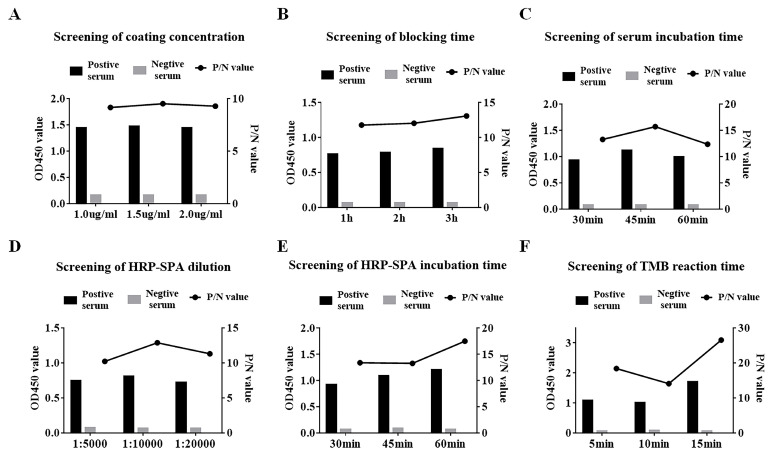
Optimization of ELISA reaction conditions. (**A**) Screening of coated antigen concentration. (**B**) Screening of blocking time. (**C**) Screening of serum incubation time. (**D**) Screening of HRP-SPA dilution. (**E**) Screening of HRP-SPA incubation time. (**F**) Screening of TMB reaction time. The left axis shows the absorbance of the data (OD_450_) and the right axis shows the ratio of the positive control to the negative control (P/N value).

**Figure 4 viruses-15-00861-f004:**
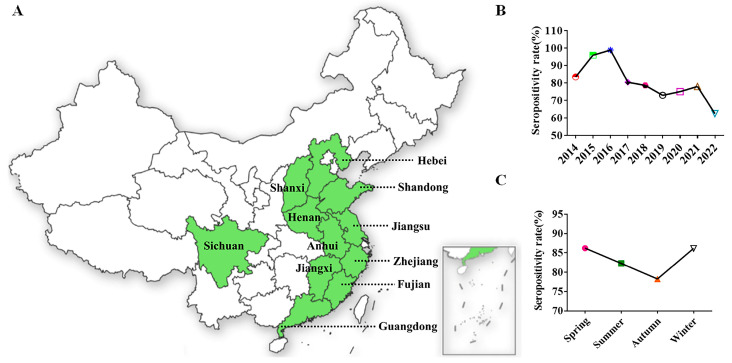
Retrospective epidemiological survey in China. (**A**) The area of light green in the map represents the sampled provinces (locations of sample collection). (**B**) SVA infection in 11 provinces from 2014 to 2022. (**C**) Seasonal epidemic distribution of SVA.

**Figure 5 viruses-15-00861-f005:**
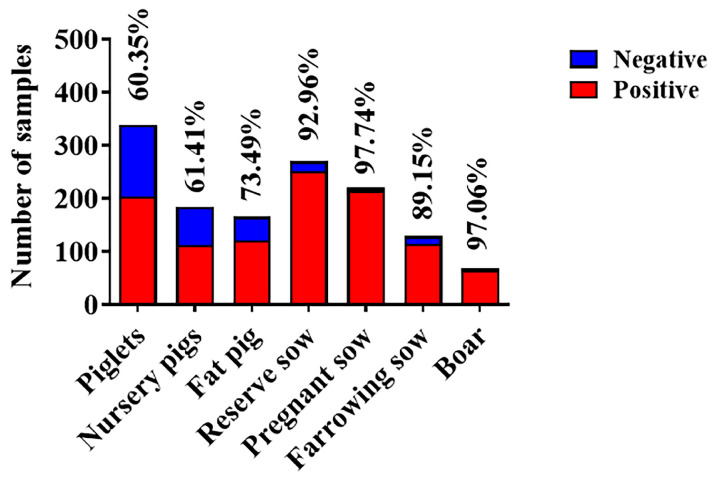
SVA infections at different growth stages. The blue column represents the amount of negative serum and the red column represents the amount of positive serum.

**Table 1 viruses-15-00861-t001:** Primer synthesis.

Primer Name	Nucleotide Sequence (5′→3′)	Restriction Sites
SVA-3AB-F	5′ GAGGAATTCAGCCCTAACGA 3′	*EcoR I*
SVA-3AB-R	5′ GCGCTCGAGTCACTGTTGCATTTC 3′	*Xho I*
SVA-VP1-F	5′ CGCGGATCCTCCACCGACAACGCCGAGACT 3′	*BamH I*
SVA-VP1-R	5′ GCTCGAGTTGCATCAGCATC 3′	*Xho l*
SVA-2C-F	5′ GAGGAATTCGGACCCATGGATA 3′	*EcoR I*
SVA-2C-R	5′ GCGCTCGAGCTGTAGAACCAGA 3′	*Xhol I*
SVA-3C-F	5′ GAGGAATTCCAGCCCAACGTGGACA 3′	*EcoR I*
SVA-3C-R	5′ GCGCTCGAGTTGCATTGTAGCC 3′	*Xho l*
SVA-3D-F	5′ GAGGAATTCGGACTGATGACTG 3′	*EcoR I*
SVA-3D-R	5′ CTCAAGCTTGTCGAACAAGGCCCTC 3′	*Hind III*
SVA-L-F	5′ GAGGAATTCCAGAACTCTAATTT 3′	*EcoR I*
SVA-L-R	5′ TATCTCGAGCTGTAGTTCGTATACGATG 3′	*Xho I*

**Table 2 viruses-15-00861-t002:** Comparison of the indirect 3AB-ELISA with VNT.

Detection Methods		VNT Results		
		Positive	Negative	Total
3AB indirect ELISA results	Positive	42	4	46
	Negative	4	50	54
Data analysis	Total	46	54	100
	Sensitivity ^1^	91.30%	-	-
	Specificity ^2^	-	92.59%	-
	Coincidence rate ^3^	-	-	92.00%

^1^ Sensitivity, the ratio of positive (42) to total (46). ^2^ Specificity, the ratio of negative (50) to total (54). ^3^ Coincidence rate, the ratio of the sum of positive (42) plus negative (50) to total (100).

**Table 3 viruses-15-00861-t003:** The overall positive rate among different provinces.

Province	Positive ^1^	Negative ^1^	Total ^1^	Ratio (%)
Shandong	730	170	900	81.11
Jiangsu	762	150	912	83.55
Jiangxi	512	119	631	73.25
Zhejiang	559	140	699	79.97
Henan	298	88	386	77.20
Hebei	125	18	143	87.41
Anhui	49	5	54	90.74
Sichuan	35	0	35	100.00
Guangdong	119	5	124	95.97
Fujian	12	5	17	70.59
Shanxi	29	0	29	100.00
Total	3230	700	3930	82.19

^1^ Number of samples.

**Table 4 viruses-15-00861-t004:** SVA infection in East China (Ratio: %).

	2014	2015	2016	2017	2018	2019	2020	2021	2022
Shandong	84.30	95.15	98.97	93.88	74.58	67.57	71.43	-	39.29
Jiangxi	96.67	96.67	99.03	82.05	80.00	74.00	75.00	66.67	37.50
Jiangsu	86.89	92.31	98.92	85.92	81.00	-	-	55.21	75.00
Zhejiang	54.24	98.67	98.60	69.88	86.67	60.26	-	-	81.55

“-” No data available.

## Data Availability

Not applicable.

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
