# Peer review of "The Establishment and Application of Indirect 3AB-ELISA for the Detection of Antibodies against Senecavirus A"

_viruses, 2023, doi:10.3390/v15040861_

Round 1

Reviewer 1 Report

Comments for the Author:

Senecavirus A (SVA) is an emerging virus affecting pigs. The development of the serodiagnosis method for SVA detection is significant in serological studies. This study established an indirect 3AB-ELISA for the detection of the antibodies against SVA. The assay was used for a nine-year (2014-2022) retrospective and prospective serological study, which showed seropositivity from 2016 (98.85%) to 2022 (62.40%). Overall, this is an advance in SVA diagnosis.

The methods and the data presented look convincing and well-presented. However, some points are needed to be addressed before publication.

I recommend the following changes:

Major issues:

1) One of the issues is that I did not see the removal of the Hig-tag after protein purification. This could cause false-positive results in iELISA due to the application of some subunit vaccines with Hig-tags. 

2) Could the authors explain the reaction band of about 50KDa and a smaller band of about 10KDa in lane 6 in the WB figure?

3) I suggest the authors use neutralizing test to confirm 20 strong positive samples detected by iELISA, whether these positive sera are false-positive or not.

Reviewer 2 Report

The strength of this paper comes from the retrospective survey, there is a commercial ELISA kit available thus reducing the novelty of the ELISA development piece. This paper needs extensive grammar and sentence restructuring as well as vocabulary changes- suggest having this manuscript edited by scientific English writing professionals.

The scientific soundness is good, however stating the value of the value of the reported findings to the swine industry should be moderated.

Round 2

Reviewer 1 Report

I have no other questions.